# Death Anxiety in Caregivers of Chronic Patients

**DOI:** 10.3390/healthcare12010107

**Published:** 2024-01-02

**Authors:** Raúl Quevedo-Blasco, Amparo Díaz-Román, Alberto Vega-García

**Affiliations:** 1Mind, Brain and Behavior Research Center (CIMCYC), University of Granada, 18011 Granada, Spain; 2Faculty of Psychology and Speech Therapy, University of Málaga, 29010 Málaga, Spain; 3Campus de Chamartín, University Alfonso X El Sabio, 28016 Madrid, Spain

**Keywords:** chronic illnesses, lethal chronic diseases, chronic disabilities, long-term care, family caregivers, informal caregivers, systematic review, meta-analysis

## Abstract

This study aimed to determine the extent to which caregivers of patients with chronic illnesses experience death anxiety, and which variables from caregivers and patients might potentially be related to their death anxiety. It also aimed to compare the levels of death anxiety between patients and caregivers. Web of Science, Scopus, PubMed, Psychology Database, Cochrane, and Google Scholar were searched for original studies available until December 2022 that quantitatively addressed death anxiety in family and informal caregivers of individuals with chronic illnesses. The methodological quality of the included studies was assessed, and a meta-analysis was conducted using Hedges’ *g* as the effect size index and the DerSimonian–Laird method to analyze differences between patients and caregivers in death anxiety. The results of the 11 included studies showed moderate levels of death anxiety in caregivers, and the meta-analysis (*k* = 7; 614 patients and 586 caregivers) revealed non-significant differences between the death anxiety experienced by patients and caregivers (pooled Hedges’ *g* = −0.03, 95% CI = −0.29 to 0.25, *p* = 0.802). Some sociodemographic and psychological factors (e.g., gender, depression, and anxiety) might be related to the death anxiety experienced, but additional research is necessary to validate these findings.

## 1. Introduction

Humans are the only known species up to this moment that has the ability to contemplate and anticipate their own death [1]. They are destined to live life knowing that, sooner or later, they will die. In this way, death becomes the only certainty in their life. However, curiously, thanatophobia or death anxiety remains a central and common theme in the human experience [2] considered as a specific phobia.

Fears of death have accompanied individuals from their beginnings, frequently appearing in art, philosophy, literature, and cultural and religious practices [3]. There is indeed evidence that humans have grappled with death anxiety since the history of our species began to be recorded [4]. Nevertheless, there is no consensus on what happens or why death induces and provokes such anxiety [5]. Furthermore, to this day, it is well-documented that there continues to be a considerable degree of ambiguity regarding the concept of death [6]. It is such a multifaceted construct, studied from such diverse perspectives, that it becomes very difficult to define. The interchangeable use of the terms anxiety and fear has also contributed to the current confusion and difficulties in its definition [7]. Despite this, it has been conceptualized to include the concern, fear, or discomfort that arises after an individual becomes aware of their own death and the death of others [7,8,9]. In summary, considering the various perspectives, death anxiety can be defined as: “*an unpleasant emotion of multidimensional concerns that is of an existential origin provoked on contemplation of death of self or others*” (p. 413; [7]).

Terror Management Theory (TMT) has been the central theoretical approach guiding research on the impact of death anxiety in a wide range of phenomena over the last four decades [3,10]. TMT proposes that innate human instincts for self-preservation, along with the awareness of our own mortality, can produce a powerful sense of fear or meaninglessness, leading to a range of maladaptive coping behaviors [11]. Some of these behaviors, such as avoidance, can be the basis for numerous mental illnesses, while others may appear unrelated to death [12].

In this regard, although TMT research has largely focused on exploring the role of death anxiety in everyday human behavior, the findings also have significant implications for mental health, where it has recently been theorized that death anxiety is a transdiagnostic construct underpinning a multitude of mental illnesses [13]. Thus, death anxiety and its related symptoms could manifest in numerous disorders such as somatic and related disorders [14], panic disorder [15], specific phobias [12], obsessive-compulsive disorder [16,17], depressive disorders [18], post-traumatic stress disorder [19], or eating disorders [20]. It is important not to overlook the consideration of factors associated with suicide [21] and the proper diagnosis of suicidal ideation and intention (see [22]).

The findings from hundreds of studies have demonstrated the robustness of this theory [23,24]. These also have implications for clinical populations [3]. Individuals facing their own mortality when diagnosed with a chronic illness may suffer from death anxiety [25,26], making it one of the most common psychological consequences of chronic illnesses [27,28]. While many of these conditions are treatable today, the myriad of physical and psychological barriers that can arise over the course of the illness can intensify death anxiety [29,30]. Additionally, many of these patients facing life-threatening traumas and conditions experience death in institutional settings [31]. Here, nurses and other healthcare professionals caring for these patients may also have death-related fears, which can negatively influence their attitudes toward providing care [32,33].

The increase in patients with chronic illnesses requiring care and advances in medicine have led institutional care to give way to home care, which inevitably increases the number of caregivers—whether they are family members, spouses, friends, or home health aides—in a home environment [34]. Moreover, many of these illnesses come with significant deterioration, necessitating that patients increasingly depend on the assistance of these caregivers [35]. Being in close contact with the patient and observing their pain and physical decline, in addition to having serious implications for the caregiver, including increased financial, emotional, and physical burdens [36,37], can also trigger their own fears of death and dying [38]. In this context of chronic illness, where it becomes more common to witness the death of a loved one or think about leaving behind family or close individuals, the impact of death anxiety on caregivers has received increasing attention since the turn of the century [39]. However, literature on the subject remains limited [40,41].

### Objectives

The main objective of this systematic review was to determine the extent to which caregivers of patients with chronic illnesses experience death anxiety, particularly in cases where the patients suffered from severe, life-threatening, or lethal chronic diseases. It also aimed to compare the death anxiety experienced by these caregivers with that experienced by the patients themselves and analyze potential variables from both the caregiver and the patient that may influence their levels of anxiety.

## 2. Materials and Methods

### 2.1. Literature Search

The systematic review (PROSPERO registration no: CRD42023487929) was conducted following the guidelines for the development and reporting of systematic reviews in Psychology and Health by [42] and the recommendations from the updated PRISMA 2020 guidelines (Preferred Reporting Items for Systematic Reviews and Meta-Analyses; [43]). A literature search was conducted in December 2022 (in accordance with some recommendations available for that purpose in the literature [44]) in the following databases: Web of Science (Core Collection), Scopus, PubMed, Psychology Database (via ProQuest, Ann Arbor, MI, USA; Clarivate), and Cochrane. Additionally, this search was supplemented with Google Scholar, and the references of the selected studies were reviewed to ensure that no research had evaded the filter. Combinations of structured keywords in English were used, replicating the search strategy: (“death anxiety” OR “fear of death” OR thanatophobia) AND (caregiver* OR spouse* OR family). The search was restricted to title, abstract, and/or keywords, with the search strategy adapted to the criteria of each database. Search limits were set for the year of publication (from 2010 to December 2022), publication type (articles and reviews), and language (English and Spanish).

### 2.2. Inclusion Criteria

Regarding the inclusion criteria, studies that addressed death anxiety in family and informal caregivers of individuals with severe or lethal chronic illnesses were selected, without restrictions on age, gender, country of origin, or religious belief. Only those studies in which the variable of death anxiety had been quantitatively measured in the population of interest and published in English or Spanish were included.

In terms of exclusion criteria, studies that did not adequately detail or investigate the psychological effects of death anxiety in caregivers were discarded. Studies in which the patients did not have chronic illnesses or in which the sample was mixed with patients with non-chronic illnesses were also excluded. Qualitative or incomplete studies, book chapters, conference abstracts, doctoral theses, and studies published in languages other than Spanish or English were rejected.

### 2.3. Selection Process

All search results from the various databases were exported to the bibliographic management software Mendeley Reference Manager for Desktop, and duplicates were manually removed. Both the initial selection of studies by title and abstract and the full-text review of the initially preselected studies were carried out by the first author of this article and an external reviewer. The results from both reviewers were compared, and discrepancies and uncertainties were resolved through consensus. The inter-rater reliability was calculated using Cohen’s kappa index.

### 2.4. Methodological Quality Assessment

The assessment of the methodological quality of the studies was conducted using the JBI Critical Appraisal Checklist for Analytical Cross-Sectional Studies by [45], with the exception of one randomized controlled trial [46] and an experimental study [47]. The former was evaluated using the JBI Critical Appraisal Tool for Assessment of Risk of Bias for Randomized Controlled Trials, as outlined by [48]. The experimental study, which investigated the impact of Rational Emotive Therapy adapted for palliative care on a sample of cancer patients and their family caregivers in Nigeria, was assessed using the CONSORT statement [49]. The criterion followed to choose one assessment tool over another was the study design of the included studies. Following the application of these assessment tools, it was confirmed that the articles included in this review exhibited medium to high quality, and thus, no documents were excluded based on quality criteria.

### 2.5. Method for Analysis and Synthesis of Information

For the data extraction from the selected articles, a template was used, which included: (a) the first author’s last name and year of publication, (b) total sample size, (c) the type of patient’s disease, (d) the country where the study was conducted, (e) study variables in the caregiver, (f) instruments used, and (g) primary results. Due to the variability of data provided by the studies, only one meta-analysis comparing death anxiety levels experienced by patients and caregivers was conducted, while the remaining study results were analyzed descriptively.

The meta-analysis was conducted using RStudio software (version 2021.09.0 + 351 for Windows). The small-sample adjusted standardized mean difference (Hedges’ *g*) was used as the effect size index, comparing the means and standard deviations of death anxiety obtained by patients and caregivers in a questionnaire or scale. The weighted effect size was calculated using the inverse variance method and the DerSimonian-Laird random-effects model, with a 95% confidence interval. Heterogeneity among the studies was assessed using the Q and *I*^2^ statistics. Sensitivity or publication bias analyses were not performed due to the limited number of selected studies.

## 3. Results

### 3.1. Study Selection

A total of 915 records were retrieved, and 192 duplicates were manually removed, resulting in the evaluation of 723 titles and abstracts for inclusion in the systematic review. In the initial screening, based on titles and abstracts, 33 studies were retained and subjected to full-text review, of which 11 were included in the systematic review. Figure 1 displays both the study selection process and the reasons for excluding studies reviewed in full text. The inter-rater reliability was excellent (*κ* = 1). The results of the methodological quality assessment indicated that the included studies did not show methodological biases.

### 3.2. Study Characteristics

The main characteristics of each study can be found in Table 1 and Table 2. The 11 included studies provided a total sample of 1836 caregivers. Among the caregivers, 747 (40.69%) were male, and 1089 (59.31%) were female. Cancer was the most common chronic illness among the patients in seven of the 11 studies, although of different types. Brain tumor, Amyotrophic Lateral Sclerosis, Multiple Sclerosis, and Acquired Immunodeficiency Syndrome (AIDS) were the chronic illnesses in the remaining four studies.

### 3.3. Death Anxiety and Related Factors

The Death Anxiety Scale (DAS; [51]) was the most commonly used instrument in the analyzed studies, with various versions employed (e.g., Thorson-Powell’s DAS [60]). Other tests used in the rest of the studies included were the BOFRETTA scale [55], the Carmel’s Fear of Death and Dying Inventory [53], the Death and Dying Distress Scale-Caregivers (DADDS-CG; [62]), and the Death Anxiety Questionnaire (DAQ; [56]). The average scores on death anxiety varied among the studies, even when using the same assessment instrument (e.g., [50,57]). See specific information for each study in Table 1.

Regarding factors associated with death anxiety, it can be observed that depression and anxiety are among the most prevalent [35], although with nuances in some cases [52]. On the positive side, the relationship with patients’ performance and dependence can also be noted, although it negatively affects their quality of life and self-control [28,46,57]. Perceived social support is also identified as an explanatory factor for greater or lower levels of death anxiety in some studies [59]. It also varies depending on the illness, as exemplified in the study of Willis et al. [61], where this anxiety was notably high in female caregivers of terminally ill patients. Lastly, the strong correlation found in one study [50], between this type of anxiety and post-traumatic growth cannot be ignored. See Table 2 for further details of each study.

Considering the demographic characteristics of the participants, the variables most related to death anxiety were gender (n = 11), age (n = 11), marital status (n = 7), educational level (n = 7), and employment status (n = 6). In this regard, female gender was the only risk factor for high death anxiety scores in the study of Alkan et al. [50]. Other aspects considered in various studies, as an example, included socioeconomic status or monthly income (n = 3 and n = 1, respectively), religion (n = 2), living environment (n = 1), or type of housing (n = 1). The most noteworthy psychological consequences are primarily found in religious caregivers [50,52]. See Table 2 for details on the characteristics analyzed in each study and the instruments employed.

### 3.4. Comparison of Death Anxiety between Patients and Caregivers

The results of the meta-analysis conducted with seven of the included studies (Figure 2) did not reveal statistically significant differences in death anxiety levels experienced by patients and caregivers (pooled Hedges’ *g* = −0.03, 95% CI = −0.29 to 0.25, *p* = 0.802). There was high heterogeneity among the studies (Q = 24.17, *p* < 0.001, *I*^2^ = 75.18%).

## 4. Discussion

The fear of death is common to the human species [63], being more evident in cases of patients diagnosed with a life-threatening or terminal illness [64], and even more so in the case of its recurrence [54]. In the case of caregivers, there are fewer studies available; therefore, this review sought to determine whether this assertion can also be applied to caregivers of patients with chronic diseases. In this regard, it is essential to consider the existing mediation models in the perceptions of caregivers and, in general, healthcare professionals (see [65]).

The DAS developed by Templer [51] over 50 years ago, remains a classic test for assessing anxiety related to existential concerns about mortality. Even today, as evidenced by the results found, despite sustained contributions to research and instrumentation in this field, it continues to be a tool used to assess attitudes towards death in the literature [66].

Considering sociodemographic factors, only in two studies were higher levels of death anxiety found in women [28,61], indicating that it is not an exclusive risk variable (see Table 2). This could be due to the fact that this relationship only occurs in very specific areas or contexts and not universally across all settings [67,68].

When caregivers confront the process of death, pain, and the fear of mortality in patients during the care process, it can impact their quality of life [69,70]. These results are consistent with those found in the analyzed studies [28,46,57,58]. An interesting finding is that, after the meta-analytical analysis of seven studies, no statistically significant differences were found in the levels of death anxiety experienced by patients and caregivers. These results provide evidence of the importance of evaluating and working in a multidisciplinary way, not only with patients but also with caregivers. Additionally, this is a guarantee of effectiveness, as the caregiver’s quality of life is essential for the well-being of the patient who requires their care and attention. To achieve this, it is crucial to take into account and alleviate, to the extent possible, the caregiver burden, especially the emotional burden [71,72].

It is important to remember that, even though this work focuses on chronic patients, it is essential to also consider caregivers of patients with other disabilities or in special circumstances. For example, children with disabilities [73] or patients in institutional settings [31], where thoughts of suicide may even arise [74], or where individuals may already have specific needs, regardless of their physical health [75]. Notably, all professionals dealing with them could be at risk of psychological issues, not only those formally recognized as caregivers, and they also require attention and support [76]. This once again emphasizes the need to consider the professional profiles most associated with suicide in general [77], as well as the different situations that healthcare professionals, in particular, have to face (e.g., [78]), and underscores the importance of exploring pathways to promote their own health [65,79].

When drawing conclusions consistent with the results obtained in this study, caution is warranted, as it is not without limitations. The examined studies are limited in number, and there are methodological variables that may affect the generalization of the data. In this regard, it would be important to have a larger volume of studies using larger, and as homogenous as possible, samples. Moreover, there is a possibility that the included information may not be complete, as only studies in Spanish or English were included, potentially omitting relevant research in other languages. Similarly, research that may be significant in this field but did not meet the inclusion criteria or is indexed in other more complementary databases or institutional repositories may not have been considered. These limitations, however, further emphasize the need to delve deeper into the study of death anxiety in caregivers, whether of chronic patients or not.

## 5. Conclusions

In summary, the studies analyzed in this systematic review highlight that, in general, caregivers of patients with chronic diseases experience significant levels of death anxiety. According to the results of our meta-analysis, these levels may not even differ from those experienced by the patients themselves. Various sociodemographic and psychological factors, such as gender, and depression and anxiety levels, could influence the extent of death anxiety experienced by these caregivers. However, many more studies are still needed to examine this issue, considering the negative impact that death anxiety has on the quality of life of both caregivers and patients.

Research on the effects of death anxiety in this population is also demanded, in order to gain a deeper understanding of the impact of these symptoms on the care that they provide to the patients with chronic illnesses. This would assist researchers and clinicians in the development and implementation of more effective psychological management tools tailored to the needs of each caregiver, taking into account both their personal variables and those associated with the patient.

## Figures and Tables

**Figure 1 healthcare-12-00107-f001:**
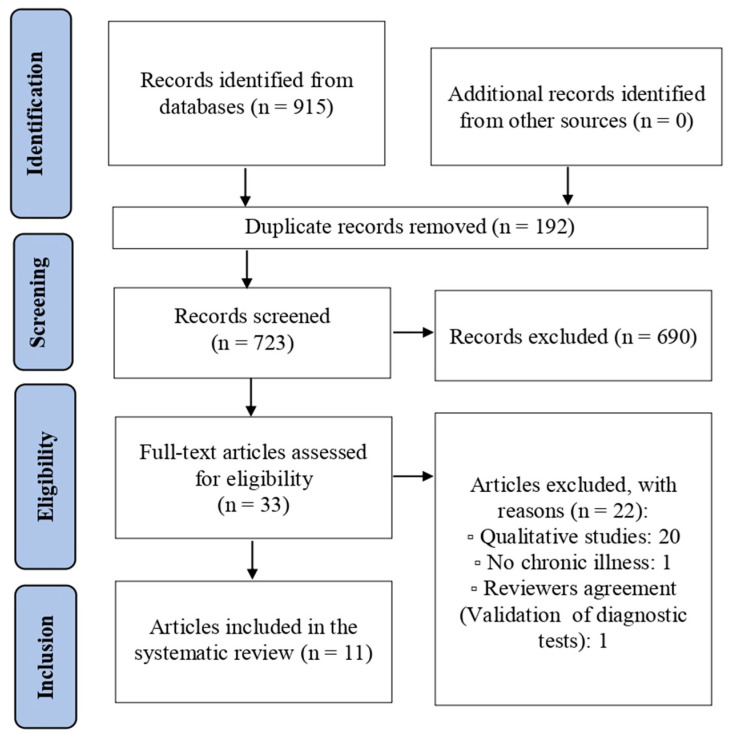
Flow Diagram of the Study Selection Process.

**Figure 2 healthcare-12-00107-f002:**
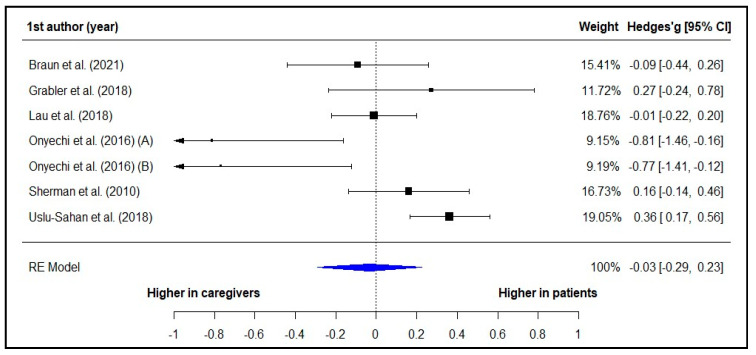
Forest Plot of the Comparison of Death Anxiety between Patients and Caregivers [35,46,47,54,58,59].

**Table 1 healthcare-12-00107-t001:** Main Characteristics and Outcomes of the Included Studies.

**Authors (Year)**	**Country**	**N (% Females)**	**Instruments**	**Results**
Alkan et al. (2020) [50]	Turkey	426 (50.23) Median age = 40.5 (17–70)	Templer’s DAS [51]	▫ Moderate anxiety in caregivers: Mean = 8 (0–15)
Bachner et al. (2011) [52]	Israel	236: 142 secular (75.35; mean age = 55.31, SD = 13.78) and 94 religious (80.85; mean age = 55.45, SD = 13.64)	Carmel’s Fear of Death and Dying Inventory [53]	▫ Low anxiety in secular caregivers: Mean = 1.89 (0–5) ▫ Medium anxiety in religious caregivers: Mean = 2.26 (0–5)
Braun et al. (2021) [54]	USA	52 (59.62) Mean age = 56.6 (34–79)	Templer’s DAS [51]	▫ Moderate anxiety in caregivers: Mean = 6.08 (0–15) ▫ No differences between patients and caregivers
Grabler et al. (2018) [35]	Germany	30 (53.33) Mean age = 57.33 (SD = 9.47)	BOFRETTA Scale [55]	▫ Moderate anxiety in caregivers: Mean = 23.5 (10–40) ▫ No differences between patients and caregivers
Lau et al. (2018) [46]	Japan	173 (53.8) Mean age = 53.2 (SD = 12.6)	Templer’s DAS [51]	▫ Moderate anxiety in caregivers: Mean = 7.65 (0–15) ▫ No differences between patients and caregivers
Onyechi et al. (2016) [47]	Nigeria	52: 26 G_T_ (84.62; mean age = 55.75, SD = 3.03) and 26 G_C_ (84.62; mean age = 55.74, SD = 3.05)	DAQ [56]	▫ High anxiety at baseline in caregivers of the GT (mean = 71; 15–75) and the GC (mean = 70.69; 15–75)
Seyedoshohadaee et al. (2019) [57]	Iran	200 (41.50) Mean age = 40.39 (SD = 11.75)	Templer’s DAS [51]	▫ Moderate anxiety in caregivers: Mean = 5.92 (0–15)
Sherman et al. (2010) [58]	USA	79: 43 AIDS (67.44; mean age = 42.4, SD = 13.6) and 36 cancer (61.11; mean age = 56, SD = 14.8)	DAQ [56]	▫ No differences in death anxiety between caregivers of AIDS and cancer patients ^a^ ▫ No correlations between scores of patients and caregivers
Soleimani et al. (2017) [28]	Iran	330 (58.79) Mean age = 40 (SD = 13.5)	Templer’s DAS [51]	▫ Moderate anxiety in caregivers: Mean = 46.7 (15–75) ^b^
Uslu-Sahan et al. (2019) [59]	Turkey	200 (71.00) Mean age = NS (55.5% < 45 years)	Thorson-Powell’s DAS [60]	▫ Moderate anxiety in caregivers: Mean = 46.83 (0–100) ▫ Higher anxiety in patients than in caregivers
Willis et al. (2022) [61]	USA	67 (73.13) Mean age = 50.61 (SD = 13.56)	DADDS-CG [62]	▫ Moderate anxiety in caregivers: Mean = 34.81 (0–75)

*Notes*. SD, standard deviation; NS, not specified; DAS, Death Anxiety Scale; G_T_, treatment group; G_C_, control group; DAQ, Death Anxiety Questionnaire; DADDS-CG, Death and Dying Distress Scale-Caregivers; AIDS, Acquired Immune Deficiency Syndrome. ^a^ Only the total sample was considered for the meta-analysis, without differentiation between the caregivers of the two patient groups, based on the data provided by the authors of the article concerning the total scale score. ^b^ The scoring range in this study (15–75) is higher than in other studies (0–15) due to the specific version of the Templer’s DAS used in this study (the Persian version).

**Table 2 healthcare-12-00107-t002:** Other Outcomes of the Included Studies.

**Authors (Year)**	**Disease**	**Variables Associated with Death Anxiety**	**Instruments**
Alkan et al. (2020) [50]	Cancer (gastrointestinal, breast, lung, prostate, gynecological, other)	(+) Posttraumatic growth, female sex, self-perception, philosophy of life, and changes in relationship	▫ Post-Traumatic Growth Inventory & Scale (PTGI) ▫ Questionnaries on demographic data and sociocultural background
Bachner et al. (2011) [52]	Cancer (NS)	(+) Depressive symptomatology in religious caregivers (−) Emotional exhaustion in secular caregivers	▫ Abridged Beck Depression Inventory (BDI) ▫ Exhaustion subscale of the Maslach Burnout Inventory (MBI)
Braun et al. (2021) [54]	Brain tumor	(+) Fear of cancer recurrence in patients and caregivers	▫ Fear of Cancer Recurrence (FCR-7)
Grabler et al. (2018) [35]	ALS	(+) Caregivers’ depression with patients’ death anxiety (+) Anxiety in caregivers, and depression in both patients and caregivers	▫ Beck Depression Inventory (BDI) ▫ State-Trait Anxiety Inventory (STAI)
Lau et al. (2018) [46]	Lung cancer	(+) Achievement and dependency in patients and caregivers (−) Quality of life in patients, and self-control in both patients and caregivers	▫ Dysfunctional Attitude Scale (DAS) ▫ Caregiver Quality of Life Index—Cancer (CQOLC)
Onyechi et al. (2016) [47]	Cancer (breast, prostate, cervix)	Beneficial effects of Rational-Emotive Hospice Care Therapy (REHCT) on death anxiety in patients and caregivers	▫ REBT Hospice Care Manual
Seyedoshohadaee et al. (2019) [57]	MS	(+) Relationship with the patient, duration of patient care, occupational status, and type of insurance (−) Quality of life	▫ Questionnaire on demographic characteristics ▫ 36-Item Short Form Health Survey (SF-36)
Sherman et al. (2010) [58]	Advanced cancer/AIDS	(−) Quality of life in patients and caregivers	▫ Quality of Life Scale (Family Version) (QLS) ▫ McGill Quality of Life Questionnaire (MQOL)
Soleimani et al. (2017) [28]	Cancer (NS)	(+) Caregiver demographic factors (being a daughter, female sex, unemployment, education level, and having family as main source of income) and patient factors (not having anticancer treatment and receiving radiation therapy) (−) Social support, socioeconomic status, engagement in social activities, prayer frequency, and quality of life	▫ Sociodemographic survey ▫ Quality of Life Scale (Family Version) (QOL-FL)
Uslu-Sahan et al. (2019) [59]	Gynecological cancer	(−) Social support in patients and caregivers	▫ Multidimensional Perceived Social Support Scale (MSPSS)
Willis et al. (2022) [61]	Brain tumor	(+) Female sex and tumor grade	▫ Reports of sociodemographic information ▫ Medical records

*Notes*. ALS, Amyotrophic Lateral Sclerosis; MS, Multiple Sclerosis; NS, not specified; AIDS, Acquired Immune Deficiency Syndrome; (+), positive relationship; (−), negative relationship.

## Data Availability

Data available on request.

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
