# Peer review of "Death Anxiety in Caregivers of Chronic Patients"

_healthcare, 2024, doi:10.3390/healthcare12010107_

Round 1
Reviewer 1 Report
Comments and Suggestions for Authors
Thank you for the opportunity to review this article. The authors aimed to evaluate the levels of death anxiety in caregivers of patients affected by chronic diseases. The topic of the paper is certainly of scientific interest. The conclusions reported by the authors confirm that this topic deserves careful attention. However, in my opinion, the paper is not adequately presented and needs improvement to be published.
My comments:
− 1. Introduction, page.1 (lines. 40-43): The definition of "anxiety" is a crucial point for understanding the article. I believe that it was not presented enough in the text. I urge the authors to rework this point, highlighting this key concept.
− 2.1 Literature Search, page.2 (line 95): the authors refer to "guidelines [44]". Considering that these are not commonly accepted guidelines, but refer to another author's product, I think the bibliographic reference can be removed.
− 2.4 Methodological Quality Assessment, page.3 (lines 126-132): Rereading this paragraph several times carefully, one understands that the authors chose different assessment tools for different articles, depending on the study design. However, with this mode of presentation, it is not clear that this was a choice made a priori, but rather a heterogeneity of evaluation methods emerges. I invite the authors to rephrase this paragraph. Clearly explaining by which criteria they decided to use different assessment tools.
− 3. Results, page.4 (lines 153-153): delete the text.
− Figure 1, page 4: the authors report that in the Article Screening process, they excluded 690 articles without specifying the reasons. I invite them to make explicit the reasons for these choices.
− 3.2 Study Characteristics, page.4 (line 168): the authors report the mean age of the total sample of 1.836 caregivers (49.9 years). I invite them also to report the standard deviation.
− Table 1, page.5: I have some suggestions
o In the column "N (% females)" I invite authors to include along with the mean age the standard deviation.
o In the line inherent in the study by Sherman et al. (2017), the authors report a total sample of 70 subjects, and later that this consists of 43 with AIDS and 36 with cancer. Since the total sample should be 79 subjects, I invite them to correct the mistake.
o In the line inherent in the study by Soulemani et al. (2017), the authors report that the assessment tool used is Templer's DAS. This about the other articles, is reported with a scale of 0-15, while only in this case is it reported as 15-75. I invite the authors to correct the mistake.
o I also believe that the Results could be presented more clearly, possibly by dividing the column into two, one containing the numerical values, and the other with the commentary.
− Table 2. page.5: considering that no others have been described above, I urge the authors to remove the word "Other" from the title of the third column
− 3.3 Death Anxiety and Related Factors, page.6 (lines 183-185): in my opinion, these lines can be deleted, as they do not add anything to the article.
− 3.3 Death Anxiety and Related Factors, page.6 (lines 187-190): I believe that in this passage, to refer to bibliographic entries, the number is sufficient (e.g. [58]). I suggest to authors to delete "see," "for example," etc.
− 3.3 Death Anxiety and Related Factors, page.7 (lines 192-194): in this part, unlike before, because the authors refer to some studies in particular, I encourage them to also use the author's name (e.g., "in the study of Willis et al. [60]")
− 3.3 Death Anxiety and Related Factors, page.7 (lines 197-201): the authors use the formula (K = X), in referring to the number of articles analyzing certain variables. I invite them to use (n = X) as done in Figure 1.
− Figure 2, page.7: in the meta-analysis, the authors differentiated the two groups included in the study by Onyechi et al, while the same was not done with Sherman et al, which also included two types of patients. I assume the reason is that, as made explicit in Table 1, in the paper by Onyechi et al the two groups of caregivers showed different levels of anxiety, whereas Sherman et al do not report different levels between caregivers of the two groups. Considering that the two groups were differentiated in Table 1, and that they are indeed two distinct populations, I would suggest that the authors differentiate them in the meta-analysis as well.
− 4. Discussion, page.8 (lines 242-245): in my opinion, this paragraph contains some references that are outside the topic of the article, including an unnecessary self-citation. I encourage the authors to remove it.
− 4. Discussion, page.8 (line 254): I invite the authors to remove "(a reduced number)" as it is repetition.
Author Response
Thank you for the opportunity to review this article. The authors aimed to evaluate the levels of death anxiety in caregivers of patients affected by chronic diseases. The topic of the paper is certainly of scientific interest. The conclusions reported by the authors confirm that this topic deserves careful attention. However, in my opinion, the paper is not adequately presented and needs improvement to be published.
Comments:
− 1. Introduction, page. 1 (lines. 40-43): The definition of "anxiety" is a crucial point for understanding the article. I believe that it was not presented enough in the text. I urge the authors to rework this point, highlighting this key concept.
Response: We thank the reviewer’s comment, and we have incorporated a definition of the term in the introduction as follows: In summary, considering the various perspectives, death anxiety can be defined as: “an unpleasant emotion of multidimensional concerns that is ofan existential origin provoked oncontemplation of death of self or others” (p. 413; [7]).
− 2.1 Literature Search, page. 2 (line 95): the authors refer to "guidelines [44]". Considering that these are not commonly accepted guidelines, but refer to another author's product, I think the bibliographic reference can be removed.
Response: We have modified the sentence referring to the reference mentioned by the reviewer to prevent the search recommendations provided in that source from being considered as commonly accepted guidelines.
− 2.4 Methodological Quality Assessment, page. 3 (lines 126-132): Rereading this paragraph several times carefully, one understands that the authors chose different assessment tools for different articles, depending on the study design. However, with this mode of presentation, it is not clear that this was a choice made a priori, but rather a heterogeneity of evaluation methods emerges. I invite the authors to rephrase this paragraph. Clearly explaining by which criteria they decided to use different assessment tools.
Response: Certainly, the criterion followed to choose one assessment tool over another was the study design of the included studies, as it is now explained in that paragraph.
− 3. Results, page. 4 (lines 153-153): delete the text.
Response: We have deleted the text.
− Figure 1, page 4: the authors report that in the Article Screening process, they excluded 690 articles without specifying the reasons. I invite them to make explicit the reasons for these choices.
Response: We appreciate the reviewer's invitation; however, only the reasons for exclusion of the articles that were reviewed in full text are provided (and not those of the articles excluded based on title and/or abstract), following PRISMA guidelines.
− 3.2 Study Characteristics, page. 4 (line 168): the authors report the mean age of the total sample of 1.836 caregivers (49.9 years). I invite them also to report the standard deviation.
Response: As mean age was not reported in one study and median age was the data reported in another one, we have decided to delete the mean overall age of 49.9 years of the manuscript to avoid possible confusions.
− Table 1, page. 5: I have some suggestions
o In the column "N (% females)" I invite authors to include along with the mean age the standard deviation.
Response: The standard deviation is now reported along with the mean age in Table 1.
o In the line inherent in the study by Sherman et al. (2017), the authors report a total sample of 70 subjects, and later that this consists of 43 with AIDS and 36 with cancer. Since the total sample should be 79 subjects, I invite them to correct the mistake.
Response: We appreciate the reviewer's observation, and we have corrected the mentioned mistake.
o In the line inherent in the study by Soulemani et al. (2017), the authors report that the assessment tool used is Templer's DAS. This about the other articles, is reported with a scale of 0-15, while only in this case is it reported as 15-75. I invite the authors to correct the mistake.
Response: The reason for that difference is now explained in the table note: “The scoring range in this study (15-75) is higher than in other studies (0-15) due to the specific version of the Templer’s DAS used in this study (the Persian version).”
o I also believe that the Results could be presented more clearly, possibly by dividing the column into two, one containing the numerical values, and the other with the commentary.
Response: Although we sincerely appreciate the reviewer's suggestion, we prefer to keep those results within the same column due to space constraints.
− Table 2. page. 5: considering that no others have been described above, I urge the authors to remove the word "Other" from the title of the third column
Response: The word “Other” was deleted.
− 3.3 Death Anxiety and Related Factors, page.6 (lines 183-185): in my opinion, these lines can be deleted, as they do not add anything to the article.
Response: The lines in question have been modified in the article in accordance with the reviewer's comment.
− 3.3 Death Anxiety and Related Factors, page.6 (lines 187-190): I believe that in this passage, to refer to bibliographic entries, the number is sufficient (e.g. [58]). I suggest to authors to delete "see," "for example," etc.
Response: Those expressions were deleted.
− 3.3 Death Anxiety and Related Factors, page. 7 (lines 192-194): in this part, unlike before, because the authors refer to some studies in particular, I encourage them to also use the author's name (e.g., "in the study of Willis et al. [60]")
Response: The author’s name has been added to every sentence “in the study of”.
− 3.3 Death Anxiety and Related Factors, page. 7 (lines 197-201): the authors use the formula (K = X), in referring to the number of articles analyzing certain variables. I invite them to use (n = X) as done in Figure 1.
Response: The formula has been changed according to the reviewer’s suggestion.
− Figure 2, page. 7: in the meta-analysis, the authors differentiated the two groups included in the study by Onyechi et al, while the same was not done with Sherman et al, which also included two types of patients. I assume the reason is that, as made explicit in Table 1, in the paper by Onyechi et al the two groups of caregivers showed different levels of anxiety, whereas Sherman et al do not report different levels between caregivers of the two groups. Considering that the two groups were differentiated in Table 1, and that they are indeed two distinct populations, I would suggest that the authors differentiate them in the meta-analysis as well.
Response: Unfortunately, the real reason for not considering both patient samples separately in the study by Sherman et al was the unavailability of the necessary data. Specifically, it was not possible to calculate the effect size for the total score on the scale with the data provided by the authors in the article. As can be observed in Table 5 of that article, the results of the Wilcoxon W tests and associated p-values are not reported for the total scale, which is the score considered in the rest of the articles. A letter was written to the first author of the article requesting the relevant data, but the authors no longer had access to them. Therefore, for the calculation of the effect size in that study, the correlation coefficient for the total scale available in Table 6 of the article was ultimately used (without separation between patient groups). We have now added a specification to that effect in Table 1 to avoid confusion: “Only the total sample was considered for the meta-analysis, without differentiation between the caregivers of the two patient groups, based on the data provided by the authors of the article concerning the total scale score.”
− 4. Discussion, page. 8 (lines 242-245): in my opinion, this paragraph contains some references that are outside the topic of the article, including an unnecessary self-citation. I encourage the authors to remove it.
Response: We have made some modifications in the Discussion and removed some citations and references in accordance with the reviewer's comments.
− 4. Discussion, page. 8 (line 254): I invite the authors to remove "(a reduced number)" as it is repetition.
Response: “A reduced number” was removed.
Reviewer 2 Report
Comments and Suggestions for Authors
The manuscript of Quevedo-Blasco et al. focuses on death anxiety among caregivers of chronically ill patients. This approach is not only interesting, but also very important and clinically relevant, especially as this group of caregivers is given little consideration in clinical work and research.
The manuscript is well structured and readable. No need for major modifications are apparent. It is to recommend that the authors should agree on a term regarding death anxiety, which they would then use consistently throughout the manuscript. To my opinion, terms such as thanatophobia and fear should be avoided. Another minor point of criticism concerns the term "chronic" diseases. Tinnitus can also be a painful chronic disease, but it certainly cannot be equated with lethal cancer or ALS. Therefore, title as well as the method section should be changed that this work is about studies with patients who suffered from mainly a severe, life-threatening or lethal chronic disease such as cancer, ALS or MS. I would like to thank the authors for this important contribution, which shows that family caregivers of severely chronically ill patients (somatic as well as mental) should come more into the focus of clinical and research interest.
Author Response
The manuscript of Quevedo-Blasco et al. focuses on death anxiety among caregivers of chronically ill patients. This approach is not only interesting, but also very important and clinically relevant, especially as this group of caregivers is given little consideration in clinical work and research.
The manuscript is well structured and readable. No need for major modifications are apparent. It is to recommend that the authors should agree on a term regarding death anxiety, which they would then use consistently throughout the manuscript. To my opinion, terms such as thanatophobia and fear should be avoided. Another minor point of criticism concerns the term "chronic" diseases. Tinnitus can also be a painful chronic disease, but it certainly cannot be equated with lethal cancer or ALS. Therefore, title as well as the method section should be changed that this work is about studies with patients who suffered from mainly a severe, life-threatening or lethal chronic disease such as cancer, ALS or MS. I would like to thank the authors for this important contribution, which shows that family caregivers of severely chronically ill patients (somatic as well as mental) should come more into the focus of clinical and research interest.
Response: We appreciate the suggestions from the reviewer and have made some modifications to the article accordingly. On the one hand, although "thanatophobia" and "fear" are terms commonly used in this context, we have now specified in the article that it is a specific phobia to better clarify the construct. On the other hand, as this fear is indeed more prevalent in severe, life-threatening, or lethal chronic diseases, this clarification has been added both in the objective and in the study's methodology.
Reviewer 3 Report
Comments and Suggestions for Authors
In the introduction, please expand on the topic of anxiety among caregivers of chronically ill patients. Please refer to research on the above topic.
The discussion needs to be improved. In the discussion, you should compare your research results with those of other authors. The current discussion is more like a summary
Author Response
In the introduction, please expand on the topic of anxiety among caregivers of chronically ill patients. Please refer to research on the above topic.
Response: Please, refer to the response to Reviewer 1 on this matter. Apart from the modifications made in this regard, we could not elaborate further due to the word limit.
The discussion needs to be improved. In the discussion, you should compare your research results with those of other authors. The current discussion is more like a summary.
Response: We deeply appreciate the reviewer's observation. Unfortunately, there are no other previous systematic reviews or meta-analyses on the same topic that we can compare our results to. This is why we were unable to follow the typical discussion structure associated with original empirical studies. Nevertheless, we have attempted to discuss our results in line with related themes, such as suicide.
Round 2
Reviewer 1 Report
Comments and Suggestions for Authors
I sincerely appreciate the authors' effort in answering my doubts.
I believe that the changes made have significantly improved the quality of the paper.